# Towards Unraveling the Histone Code by Fragment Blind Docking

**DOI:** 10.3390/ijms20020422

**Published:** 2019-01-19

**Authors:** Mónika Bálint, István Horváth, Nikolett Mészáros, Csaba Hetényi

**Affiliations:** 1Department of Pharmacology and Pharmacotherapy, Medical School, University of Pécs, Szigeti út 12, 7624 Pécs, Hungary; monibalint18@gmail.com; 2Chemistry Doctoral School, University of Szeged, Dugonics tér 13, 6720 Szeged, Hungary; horvathi@gmx.de; 3Department of Biochemistry, Eötvös Loránd University, Pázmány Péter sétány 1/C, 1117 Budapest, Hungary; eperke93@gmail.com

**Keywords:** peptide, interaction, translation, methylation, target, ligand

## Abstract

Histones serve as protein spools for winding the DNA in the nucleosome. High variability of their post-translational modifications result in a unique code system often responsible for the pathomechanisms of epigenetics-based diseases. Decoding is performed by reader proteins via complex formation with the N-terminal peptide tails of histones. Determination of structures of histone-reader complexes would be a key to unravel the histone code and the design of new drugs. However, the large number of possible histone complex variations imposes a true challenge for experimental structure determination techniques. Calculation of such complexes is difficult due to considerable size and flexibility of peptides and the shallow binding surfaces of the readers. Moreover, location of the binding sites is often unknown, which requires a blind docking search over the entire surface of the target protein. To accelerate the work in this field, a new approach is presented for prediction of the structure of histone H3 peptide tails docked to their targets. Using a fragmenting protocol and a systematic blind docking method, a collection of well-positioned fragments of the H3 peptide is produced. After linking the fragments, reconstitution of anchoring regions of the target-bound H3 peptide conformations was possible. As a first attempt of combination of blind and fragment docking approaches, our new method is named fragment blind docking (FBD).

## 1. Introduction

In the past decades, epigenetics has opened up new pathways of drug discovery [1]. Among epigenetic events, post-translational modifications (PTM) of histone proteins are of particular interest [2,3,4]. The epigenetic role of these PTMs can be explained by their altering effect on the chromatin structure influencing histone–DNA and histone–histone contacts in nucleosomes of chromatin fibers. Histones are small, conserved eukaryotic proteins, with a very flexible N-terminal tail (Figure 1A) composed of ca. thirty-six amino acids [5] at histone H3. The tail can be covalently modified at side-chains of amino acids K, R, T, and S. The resulted PTMs may have diverse chemical composition such as methylation, acetylation, phosphorylation, etc. If considering methylation as an example, three (mono-, di-, and tri-methylated) PTMs can be derived by replacing hydrogen atoms of the charged amino group of the side-chain of K. The N-terminal histone H3 peptide has seven locations of K (Figure 1A) and methylation can result in 4^7^ PTM variations (four comes from the three PTMs plus the non-modified K). In this way, an enormous number of PTM variations can be derived if all above-mentioned amino acids and modifying groups are considered. 

Beyond the genetic code, a “histone code” was proposed [6] based on this large number of PTM variations. The histone code is fundamental in the epigenetics of chromatin-related pathomechanisms of various diseases [2,3] and can be “decoded and translated” by chromatin-associated reader proteins [5,6,7]. Atomic resolution structures of histone-reader (writer) complexes are keys of unraveling the histone code and drug design. The large number of PTM variations of histones yields a similarly large number of possible complexes. Large scale structure determination of such complexes is challenging even for high throughput crystallographic [8,9] techniques. 

To answer the challenge, the use of theoretical approaches would be an alternative to experimental techniques. Computational docking [10,11,12] of the histone peptide to the reader (writer) target would be an obvious theoretical approach in this case. However, there are practical problems with the use of this approach. First of all, the whole N-terminal peptide tail is too large for fast docking approaches [13] due to its large torsional flexibility resulting in a complicated search problem [14]. Secondly, even the approximate location of binding sites of the histone peptide on the target surface is unknown in many cases. Approximate, pre-docking location of the binding sites is further encumbered by the shallow binding surfaces of reader proteins without deep pockets.

Docking of fragments instead of the whole histone peptide would tackle the first problem. Fragment-based approaches [15] have been used in past studies with success. In the case of peptide docking, fragmenting will result in a reduced degree of torsional freedom, and a relieved search. The second problem can be answered by the blind docking approach [16,17,18] which scans the entire surface of the target molecule without prior knowledge of the binding site. In a recent paper, a systematic blind docking method was released [19] for finding all possible binding modes of a ligand on a target.

In the present study, a combination of the fragment and systematic blind docking approaches is introduced and tested for complexes of histone peptides with their targets. The resulted methodology is named after the parent techniques as fragment blind docking (FBD).

## 2. Results and Discussion

A set of five histone H3 peptide–target complexes (Table 1) was used for elaboration and test of FBD. The complexes contain both methylated (3qla, 5tdw) and non-methylated (2ke1, 2pvc, 4lk9) histone H3 tails. Experimental complex structures deposited in the Protein Databank (PDB, [20]) mostly include deca-peptide-sized sections of the tail [21]. Seemingly, it is challenging to capture all 36 amino acids of the tail for current structure determination techniques such as X-ray or NMR. This can be explained by the mobility, and by the lack of or weak interactions of the C-terminal end of the tail with the target protein which will be further analysed in the next paragraphs. The above experimental difficulties and need for of determination of the histone-target complexes motivated the elaboration of FBD. In the forthcoming Sections, the main steps (Figure 1) of FBD are introduced using the structure of H3 peptide in complex with autoimmune regulator protein plant homeodomain (System 2ke1) as an example (Figure 1B).

### 2.1. Fragmenting

As the name of FBD indicates, the protocol (Figure 2) is based on the splitting of the original histone H3 peptide ligand (H3) into fragments. The fragments have smaller size, and fewer active torsions than the original peptide. Due to the decrease of their overall freedom, they are expected to impose less challenge on the docking search algorithm which is the rationale behind the fragmenting approaches. However, as fragment docking has been used in focused mode (see the Introduction for references), it is not clear how small fragments give the best possible results in a blind search over the entire surface of the target. To answer this question, fragments of H3 were designed according to a systematic scheme (Figure 3). The size of fragments ranged between di- and tetra-peptides. Notably, we did not use fragments larger than tetra-peptides as the above benefits of fragmenting would diminish beyond this size [12]. According to the starting position of fragmenting, two series of fragments were produced and named as N- and C-terminal fragments. To avoid single amino acid fragments, dipeptide fragmentation was carried out for the cases of tri- or tetrapeptide fragmenting, if the remaining C- or N terminal sequence were either a tetra- or pentapetides, respectively.

This fragmenting scheme is systematic and at the same time diversifies the sets of fragments used for docking. The original H3 peptide was cut at the amide bond between the carbon and the nitrogen atoms. In general, acetyl (Ac-) and imino-methyl (-NHMe) groups are used to cap/block the free N- and C-terminal cut ends of the peptides to mimic the backbone. In the case of the example system 2ke1, fragmenting resulted in thirteen different peptides (Figure 3). The structures of all fragments were prepared (Methods) and collected in a library for the blind docking cycles of the Wrapping step.

### 2.2. Wrapping

The next step of FBD involves the Wrapper module of a new method [19] which covers the entire surface of the target molecule by a mono-layer of the copies of a peptide ligand using a series of blind docking cycles. Wrapping the target into ligand copies allows systematic mapping of all possible binding modes of a ligand [19]. In the present study, wrapping of target proteins of all test systems were performed by all fragments. In the case of 2ke1, blind docking of 13 unique fragments yielded more than nine thousand docked ligand conformations (see Methods) which was distilled into 529 binding modes (binding sites and conformations) used in the next, linking step. All binding modes were compared to the corresponding experimental conformation of the full H3 peptide using a standard procedure as implemented in the Wrapper module [19]. Briefly, a root mean squared deviation (RMSD, Equation 1) value was calculated for the binding modes obtained in a wrapping cycle, and the best RMSD values were collected for each peptide fragment (Appendix A). An abridged version of the results is shown in Table 2 with binding modes of an RMSD ≤ 4 Å. This agreement with the experimental conformation was found appropriate for the linking step (Figure 2) described in the next Section. Serial number of the wrapping cycle, and the energy rank (see Methods for ranking details) holding the binding mode are also indicated in Table 2 and Appendix A. The results of the fragmenting-wrapping tests of our systems (Table 1) allowed investigation of various factors influencing FBD, such as fragment size, chemistry of binding, secondary structure of the ligand and fragment ends as described in the following paragraphs.

### 2.3. Fragment Size

It can be observed that in most of the cases, dipeptide fragments were selected for Table 2. Among the dipeptide fragments, docking of the AR-NHMe fragment was the most successful at Systems 2ke1, 4lk9, and 5tdw. Excellent RMSD values were obtained ranging from 1.7 to 2.0 Å. In these three cases the structure with the best RMSD was found in the first wrapping cycle, ranking in the top five cluster representative. The R_2_ residue of this dipeptide seems to be an important anchoring residue of the H3 histone tail. For an in-depth analysis of target–ligand interactions, the number of intermolecular contacts (N_inter_) and per-residue intermolecular interaction energy values (E_inter_) were calculated for the energy-minimized complex structures of Table 1 as described in Methods. The results of the analysis (Figure 4, Appendix A) underline that R_2_ is indeed an important anchoring residue having the largest N_inter_ and best E_inter_ values. For system 3qla, a mis-docked AR-NHMe (Appendix A) was produced due to the lack of negatively charged residues and the targeted ATRX-DNMT3-DNMT3L (ATRX-ADD) domain side resulting in a single contact at R_2_ (Figure 4, Appendix A). In case of 2pvc, the AR-NHMe fragment found the binding conformation at an RMSD of 3.2 Å in the second cycle of wrapping. This is due to the relatively small E_inter_ of R_2_, if compared to that of K4 which is the main anchoring residue (Figure 4). The importance of K_4_ is also reflected by its successful docking when K_4_ was part of the fragmented sequence (N2, N3 and C2 in Table 2).

Previous docking studies claimed [12] that appropriately docked conformations can be obtained for small peptides with low number of active torsions whereas large peptides are true challenges for most of the docking procedures [12,22,23]. In agreement with these previous reports, low RMSD values were obtained mostly for dipeptide fragments of the N2 or C2 series (Table 2). Depending on the starting point of fragmenting (C- or N-terminus) different sets of peptides could be obtained. Plausibly, identical fragment sets can be obtained if, e.g., dipeptide fragments are produced from a H3 peptide of even number of residues (Figure 3). However, the N-terminal fragmenting was successful for any fragment lengths as the anchoring residues are close to the N-terminus of H3. In the case of C-terminal fragmenting, the first few fragments often bind weakly (Figure 4, System 4lk9) to the target protein, or in some cases they are completely immersed in the bulk solvent (Figure 4, System 5tdw). At the same time, the last fragments are usually larger peptides (tri- or tetra-peptides) which makes it more difficult to find the binding conformations known from the reference structure, as also explained above.

In some cases, tripeptide (2ke1), and tetra-peptide (2pvc) fragments were also docked at good RMSD values of 2–3 in the first ranks (Table 2). Therefore, tri- and tetra-peptides can also be useful in FBD for other systems. However, dipeptides performed overall better than larger ones for (partial) reconstruction of the original H3 ligand.

### 2.4. Chemistry of Binding

The investigated H3 peptide sequences are identical in all five Systems (with variable lengths) and unmodified in three Systems (Table 1). In the cases of 3qla and 5tdw, tri-methylation occurs at K4 and K9, respectively. Such PTMs are key elements of the histone code as it was discussed in the Introduction. The investigated target proteins were different in the five Systems. Identity of the H3 ligand sequences offers the possibility of observation and analysis of differences on the interacting target side. The K_4_ binding site is negatively charged due to the presence of E (2ke1, 4lk9) and/or D (2ke1, 2pvc, 3qla) residues. K_4_ found its position correctly, except for 5tdw, where the binding site is composed of T and F accommodating the tri-methylated K_4_ side-chain in the reference crystallographic structure. At the same time, H3 fragment with the same PTM at K_9_ docked correctly (Table 2) to its negatively charged pocket with E, Q and also hydrophobic (Y) residues.

In three (2ke1, 2pvc, and 4lk9) cases, the first two dipeptide fragments are the best docked, which is somewhat expected due to the anchoring role of both R2 and K4 residues. In Figure 4, it can be observed that anchoring (see dotted line in Figure 4) of the peptide sequence is achieved by mainly two residues, R_2_ and K_4_, which can explain the positive docking results obtained especially for the fragments containing these residues (Table 2). More than that, the interaction energies calculated for these residues are also the strongest in the H3 histone tail (see dotted line in Figure 4). In case of 3qla the anchoring was also achieved by the tri-methylated K_9_ (Figure 4).

In cases of 2ke1, 2pvc, and 4lk9, consecutive fragments (AR-NHMe, Ac-TK-NHMe) have excellent RMSD ≤ 4 Å. Beside the good match with the reference structure, the distance between C_T_ and N_T_ atoms (Figure 5), in these consecutive segments fall below 0.75 Å, set as a criterion for selection of linkable ligand binding modes (see Section Linking). Hence, these ligand-binding modes are suitable for covalent re-linking.

Notably, the binding modes with the above-mentioned excellent RMSD values (Systems 2ke1, 4lk9, and 5tdw) were produced in the first wrapping cycle, i.e., one hundred blind docking runs (see Methods) were enough to identify them. In the case of System 2pvc, the best binding mode of an RMSD of 3.2 Å was found in the second cycle of wrapping, requiring an additional cycle of one hundred runs. This exemplifies the systematic approach of Wrapper which surely finds a binding mode in higher cycles even if it was not identified in the first one.

### 2.5. Secondary Structure of the Ligand.

Target-bound H3 histone peptides of the tests systems adopted a variety of secondary structures, such as coil (2pvc, 5tdw), β-sheet (2ke1, 3qla) and α-helix (4lk9). In the case of 4lk9, due to the α-helical secondary structure only the first two dipeptide sequences were found successfully with 2.9 and 4.0 Å RMSD, respectively (N2 series, Table 2). In the second fragment, the Ac-TK-NHMe peptide, is part of the α-helix in the original 4lk9 structure, resulting in the increase of RMSD if compared with that of the first fragment. Similarly, further fragments from the α-helical region could not find the reference conformation below 4 Å RMSD. Fast docking approaches such as AutoDock 4.2 often have difficulties in reproducing helical secondary structures of peptides. This is probably due to missing explicit solvent model and inadequate consideration of intra-backbone H-bridges. Notably, the scoring function of fast docking methods are trained primarily [24] for optimization of intermolecular (target–ligand) interactions and not intramolecular ones. For this reason, in AutoDock 4.2 there is an option for restraining backbone torsions of the ligand [25]. However, as a real test, we aimed at fully flexible blind docking of fragments without using any prior knowledge of their bound conformation. Finding the C-terminal region of the H3 peptide tail was difficult in all of the test cases indifferent of the secondary structure, due to the shallow binding site and weak interaction with the target protein, as reflected by the calculated E_inter_ and N_inter_ values (Figure 4), as well. However, in the case of our 4lk9 test system, where the H3 peptide tail has an α-helix secondary structure when bound to the target protein, the intramolecular hydrogen bonds in the helix made the docking even more challenging than in other test cases.

### 2.6. Fragment Ends

The fragmenting method of FBD allowed to check the role of terminal end groups (Figure 3) of peptide ligands. The AR sequence appears at two different positions of the H3 peptide (Table 1). Thus, two dipeptide fragments were formed, one with a free, positively charged and another one with a capped N-terminal. Docking results showed that the positive charge is essential to find the reference binding position at an RMSD of 1.9 Å RMSD (due to a hydrogen bond with P331 (Appendix A) carbonyl oxygen of the autoimmune regulator (AIRE) protein. In the case of the capped (Ac-AR-NHMe) version, this interaction could not occur due to lack of the positive charge. This example hints that for appropriate modelling of terminal fragments, the original (charged) end should be retained. Capping should be used only at the cleaved amide bonds as it was indicated in Section Fragmenting.

### 2.7. Linking

Having all binding modes of all fragments produced in the previous steps, pairwise linking of fragments of the same length is performed automatically by an algorithm elaborated for FBD and described in the following paragraphs. The algorithm probes all possible pairwise combinations and produces the longest possible peptide from the connected amino acid pairs. The work flow of the algorithm (Figure 6) is described for the example of the N-terminal hexa-peptide ARTKQT of histone H3. Let us suppose that this hexa-peptide was cleaved into three dipeptide fragments AR, TK, and QT, as it was described in Section Fragmenting. For FBD, the dipeptides were blocked with Ac- and/or -NHMe groups at the cleavage sites. The linker algorithm takes the first dipeptide pair of AR-NHMe of n_1_ binding modes and Ac-TK-NHMe of n_2_ binding modes, and removes the blocking groups. In this way, two series of free radicals are obtained with terminal carbon (C_T_) and nitrogen (N_T_, Figure 5) atoms available for re-forming the amide bonds. However, not all fragment binding modes are in a correct position to allow the formation of the amide bond. In some cases, the distance between C_T_ and N_T_ (d_CN_) is too large to allow re-formation of a covalent bond. To select the copies with appropriate distances, all (n1 × n2) d_CN_ values are calculated and saved in a matrix (Table 3). The d_CN_ values are generated for all dipeptide fragment pairs, and therefore, two d_CN_ matrices are produced in the case of our hexa-peptide example. There is a user-defined minimal distance tolerance d_CN,min_ = 0.75 Å comparable to the half-length of a C_T_–N_T_ bond in an amide group. A collision is identified between C_T_ and N_T_ if the actual d_CN_ ≤ d_CN,min_ and the corresponding element of the collision matrix (c_CN_) is set to zero (otherwise one). A maximal distance tolerance (d_CN,max_) of 6 Å is also defined to exclude fragment pairs too far from each-other, and a trimming matrix t_CN_ is generated based on this value. Notably, d_CN,max_ should not be too large, it must have an meaningful physical value.

To avoid overall collisions between any atoms of the fragment pairs, a distance matrix d_all_ with pair-wise distances (Table 3) between all heavy atoms is also generated. A collision matrix c_all_ is also calculated from d_all_ to identify the steric collisions between heavy atoms of the fragments. A collision is identified if the actual d_all_ ≤ d_all,min_, where d_all,min_ is a user-defined minimal distance tolerance, and a d_all,min_ = 0.75 Å, the same value as the above d_CN,min_ works well for peptide ligands. The elements of c_all_ are set to zero by default, and one if there is no collision between a pair of atoms. Finally, a filtering matrix f_CN_ is produced which tells if a fragment pair can be considered for welding and refinement (see next Section). Values of f_CN_ is set to one if t_CN_ and c_CN_ are equal to one otherwise zero.

After sorting the elements of the distance matrix in an increasing order of d_CN_ the main loop selects the first (next), unchecked fragment pair with the actual smallest d_CN_ from the list and checks the corresponding element of the filtering matrix. If the f_CN_ = 0 then the next fragment pair will be checked, otherwise the structure of the actual fragment pair is saved to the candidate pool containing structures of possible fragment pairs in separate directories.

After producing a pool of candidate structures of the first fragment pair of AR and TK in our example, the same procedure is repeated for the next fragment pair of TK and QT if considering our example hexa-peptide ARTKQT. Having all fragment pairs (two pairs in our example) finished, the candidate pools are further processed to link the fragment pairs into triads. Accordingly, the first fragment pair AR-TK with the smallest d_CN_ is selected. If the same TK fragment copy occurs in one of the TK-QT pairs, the structure of the hexa-peptide ARTKQT is produced (Figure 3, AR_2_-TK_3_-QT_1_ or AR_2_-TK_3_-QT_4_, etc.). If not, then the next AR-TK pair will be checked for a common TK with the TK-QT pairs and the algorithm proceeds until all AR-TK pairs are checked. The linker produces structure pools at all levels of the above pairing process. The process works on arbitrary long peptide chains. That is, pools of fragment pairs, triads, and tetrads are produced depending on the length of the actual peptide. A statistics of the pools is written into a report file (Appendix A).

### 2.8. Welding and Refinement.

The pools of linked candidate structures are further processed to re-form (weld) the covalent bonds between atoms C_T_ and N_T_ between fragments AR and TK in our example (Figure 5). During welding, AR and TK are rotated along the C_α_–C_Τ_ (angle Ψ) and N_T_–C_α_ (angle Φ) bonds, respectively. Rotations are performed systematically with a step size of 1 degree. One rotation step at angle Ψ is followed by a series of steps of a complete turn-around Φ, up to 360 degrees. After each rotation step, d_CN_ is calculated and stored with the corresponding angles. From among the stored d_CN_ values, the smallest one is selected and the corresponding structure is resulted by welding. The same welding procedure is followed for the remaining fragments of ARTKQTARKS presented in the linked candidate pool.

Following the linking and welding processes, structural refinement of the paired fragments is also recommended using a common molecular mechanics energy-minimization, preferable in explicit water model (Methods). In case of System 2ke1, docking found the first two di-peptide fragments at RMSD values of 1.9 (AR-NHMe) and 2.8 (Ac-TK-NHMe) Å, respectively (Table 2). These two fragments can also be identified as AR04 and TK01 in Appendix A. These fragments obtained from docking were linked and welded as it was described above and also shown in Figure 7. After refinement of the welded fragment pairs (Figure 7), the d_CN_ of the amide bond changed from 1.5 to 1.3 Å. The optimized structure of ARTK, matches the X-ray structure at a 1.3 Å RMSD. This is a remarkable improvement, considering the above-mentioned RMSD of the di-peptide fragments alone.

## 3. Methods

### 3.1. Wrapping

#### 3.1.1. Preparation of Targets

Target structures were obtained from the Protein Databank (PDB) entries of the complexes as listed in Table 1. In case of missing atoms of the amino acids, Swiss-PdbViewer was used to complete the structure [26]. Water and ion molecules were removed from the target structure. Prior to docking, energy minimization was carried out on 2pvc, 3qla, 4lk9, 5tdw. A two-step energy minimization was done using Gromacs 5.0.6 [27]. In the first step, a steepest descent was performed, followed by conjugated gradient. The structure optimization was done in AMBER99SB-ILDN force field [28] with TIP3P explicit water model [29]. The target structure was placed in the centre of a cubic box. Distance between the box and the solute atoms was set to 10 Å. The simulation box was filled with water molecules and counter-ions in order to neutralize the total charge of the system. The Particle Mesh Ewald method was used for long-range electrostatics. The van der Waals and Coulomb cut-offs were set to 11 Å. Convergence threshold of the first step (steepest descent) was set to 10^3^ kJ mol^−1^ nm^−2^, in the second step (conjugant gradient) minimization it was set to 10 kJ mol^−1^ nm^−2^. Position restraints were applied on the heavy atoms with a force constant of 10^3^ kJ mol^−1^ nm^−2^ during the energy minimizations.

#### 3.1.2. Preparation of Ligands

The non-modified peptide sequences were built using Tinker program package [30], with the protein, newton and xyzpdb commands. The optimization of the constructed ligand structures was performed using the Amber99 force field [31]. A 10^−4^ kcal/mol gradient was set to the newton program for minimization. Methylated peptide sequences were prepared with Schrödinger Maestro program package [32] by capping of the N- and C-terminal of the fragmented regions, and by adding the hydrogens. The obtained ligand structures, were optimized by Open Babel [33] using a steepest descent optimization, with 10^4^ steps. The next step was a conjugate gradient minimization, with a maximum of 10^4^ steps, and the convergence threshold was set to 10^−7^ kcal mol^−1^ Å^−1^. MMFF94 force field [34] was used in both steps. The minimized target and ligand structures were used as inputs for docking, after preparation with AutoDock Tools 1.5.7 [35]. Gasteiger–Marsili [36] partial charges were added for both, the minimized ligand and target atoms as well. All default active torsions are kept for the ligand, but the target is treated rigidly, without active torsions.

#### 3.1.3. Grid maps and Blind Docking Parameters

The grid boxes were generated around the entire protein target with AutoGrid 4.2 [25]. Grid boxes were automatically centred on the target, and grid maps of 200 grid points along all axes, with 0.375 Å spacing were generated. The AutoDock 4.2. [25] program package was used with Lamarckian Genetic Algorithm (LGA), AutoGrid 4.2 was used for calculation of grid maps of the target molecule with pre-calculated energy values. One hundred BD runs were performed, in each cycle, with 20 million maximum number of energy evaluations and the docked ligand structures are collected in a log file. Docking parameters were used as described in the previous study [18].

#### 3.1.4. Wrapping Cycles

For each BD cycle, 100 docked ligand copies were generated [17]. Docked ligand conformations were clustered and ranked based on their intermolecular energy (E_AD4_), calculated by the AutoDock 4.2 scoring function (1st energy component of estimated free energy of binding in the log file), and closest distance between each heavy atom of the ligand copies (d_min_). In the initial clustering phase, the 100 docked ligand conformations were sorted according to the E_AD4._ Ligand conformation of the lowest E_AD4_ from among the 100 docked ligand copies were selected as the representative of Cluster 1. Ligand conformation of the 2nd lowest E_AD4_ was selected as a representative of a new Cluster 2 if d_min_>drnk, where drnk is a ranking tolerance, a measure of separation of clusters from each other. This wrapping method achieves the coverage of target surface with a monolayer of N ligand copies ending up in a target–ligand_N_ complex. Ligand copies and interacting target surface elements are excluded from successive BD cycles via assignation of “excluded” atom type as detailed in our previous publication [19]. Further details on structural and physical chemistry of the Wrapper algorithm can also be found in the original publication of Wrap ‘n’ Shake [19].

After the complete coverage of the target surface, a trimming was performed, where excess ligand copies not interacting with the target were removed after the final cycle and the results were written into a single PDB file. This trimmed PDB file was further used to link the obtained ligand copies of the fragmented segments.

Root mean squared deviation (RMSD) between the calculated (C) and the experimental reference (R) ligand conformations was calculated according to Equation (1).
(1)$$RMSD=\sqrt{\frac{1}{NH_{L}}\overset{NH_{L}}{\underset{i=1}{\sum }}|C_{i}-R_{i}{|}^{2}}$$
where, NH_L_ is the number of ligand heavy atoms, R is the space vector of the i^th^ heavy atom of the experimental reference ligand molecule, C is the space vector of the i^th^ heavy atom of the calculated ligand conformation as resulted from docking.

### 3.2. Analysis of Target–Ligand Interactions

#### 3.2.1. Preparation of the Target–Ligand Complexes

The X-ray structure of the target–ligand complexes were subjected to structure optimization, using the same GROMACS and force field parameters as detailed in Section Wrapping (Preparation of targets). The only exception from the above-mentioned protocol, was the use of position restraints on the heavy atoms with a force constant of 10^4^ kJmol^−1^ nm^−2^ during the energy minimizations.

#### 3.2.2. Number of Intermolecular Interactions (N_inter_)

Target residues with a closest atomic distance below 3.5 Å measured from the H3 peptide ligand were collected and counted. Only heavy atoms were considered during the distance calculations. The list of interacting target residues can be found in Appendix A, and N_inter_ values are presented in Figure 5.

#### 3.2.3. Target–Ligand Intermolecular Interaction Energy (E_inter_)

The energy-minimized target–ligand complexes were also subjected to calculation of intermolecular interaction energies expressed as the sum of Lennard–Jones (LJ) and screened Coulomb potentials [37] (Equation (2)). For both the LJ and Coulomb potentials, Amber99sb-ildn force field parameters were used [28].
(2)$$E_{\mathrm{inter}}=\overset{N_{T}N_{L}}{\underset{i,j}{\sum }}\left[\frac{A_{\mathrm{ij}}}{r_{\mathrm{ij}}^{12}}-\;\frac{B_{\mathrm{ij}}}{r_{\mathrm{ij}}^{6}}+\;\frac{q_{i}q_{j}}{4{\pi \varepsilon }_{0\;}\varepsilon_{r}r_{\mathrm{ij}}}\right]\;A_{\mathrm{ij}}={\;\varepsilon }_{\mathrm{ij}}R_{\mathrm{ij}}^{12}\;B_{\mathrm{ij}}=\;2\varepsilon_{\mathrm{ij}}R_{\mathrm{ij}}^{6}\;R_{\mathrm{ij}}={\;R}_{i}+{\;R}_{j}\;\varepsilon_{\mathrm{ij}}=\;\sqrt{\varepsilon_{i\;}\varepsilon_{j}}\;\varepsilon_{r}=C+\;\frac{D}{1+{\mathrm{ke}}^{-{\lambda \mathrm{Br}}_{\mathrm{ij}}}}\;C\;=\;\varepsilon_{0}\text{ – }D;\;\varepsilon_{0}\;=\;78.4;\;D\;=\;-8.5525;\;k\;=\;7.7839;\;\lambda \;=\;0.003627$$
where, ε_ij_ is the potential well depth at equilibrium between the i^th^ (ligand) and j^th^ (target) atoms;

ε_0_ is the dielectric constant of bulk water at 25 °C; R_ij_ is the inter-nuclear distance at equilibrium between i^th^ (ligand) and j^th^ (target) atoms; q is the partial charge of an atom, used in AMBER99SB-ILDN force field; r_ij_ is the actual distance between the i^th^ (ligand) and j^th^ (target) atoms; N_T_ is the number of target atoms; N_L_ is the number of ligand atoms.

### 3.3. Linking and Welding

Algorithms of linking and welding were scripted in java using JDK version 1.8. into a single code FragmentMerge. The script can be run as described in Appendix A and uses a set of PDB files as resulted by Wrapping. There is also an input text file containing the name of the system (for report purposes) and the fragments line by line (Appendix A). The algorithm uses a class hierarchy. The fragments, the atoms and the bonds in the fragments, the PDB files have own class to represent them. For the rotations of welding, the atoms are stored in a molecule graph in the memory, it helps to calculate the new coordinates during the rotation process. The outputs are saved in a separate directory for pairs, triads, tetrads, pentads, etc. They are PDB files including the linked fragments. A report file after the linking and welding process contains all information about inputs, outputs, parameters and access paths.

The number of the fragment and the name of the input files are listed in REMARK lines and the name of the files refers to the content. The welding algorithm also needs connection information between the atoms. For this, coordinate files are converted by Open Babel [33] into PDB with connectivity lists.

## 4. Conclusions

In the present study, a new method, FBD, was introduced and tested on the examples of complexes of reader and writer proteins with histone H3 peptide fragments. Heuristic search engines of present fast docking methods cannot handle peptide ligands with numerous internal rotations [12,19]. The large size and flexibility of peptide ligands together with the shallow binding surface of the targeted proteins impose a big challenge on experimental structure determination methods, as well. Moreover, interaction of the C-terminal section of the histone peptides with their targets is often weak and even not visible in the experimentally determined structures. We showed that fragmenting the ligands into small peptides provide reasonable solutions even if the entire protein surface was targeted in blind docking runs. Notably, fragmenting has been used in previous fast docking studies focusing on a known binding pocket. The present study provided the first application of fragmenting in a blind docking context with no restriction of the search space. Thus, even the approximate knowledge of location of binding pocket was not necessary in our successful examples. Despite the above challenges, N-terminal anchoring fragments were correctly positioned and linked using the results of our systematic blind docking search (Wrapper). All-in-all, FBD benefited from the philosophy of its parent methods, fragment and blind docking. Present limitations and mis-docked examples of FBD come from the simplified docking force field and the lack of an explicit water model. However, these limitations can be improved by molecular dynamics simulations in many cases as it was described previously [19]. The systematic approach of FBD will improve the efficiency of structure determination of problematic complexes with large ligands such as histone peptides.

## Figures and Tables

**Figure 1 ijms-20-00422-f001:**
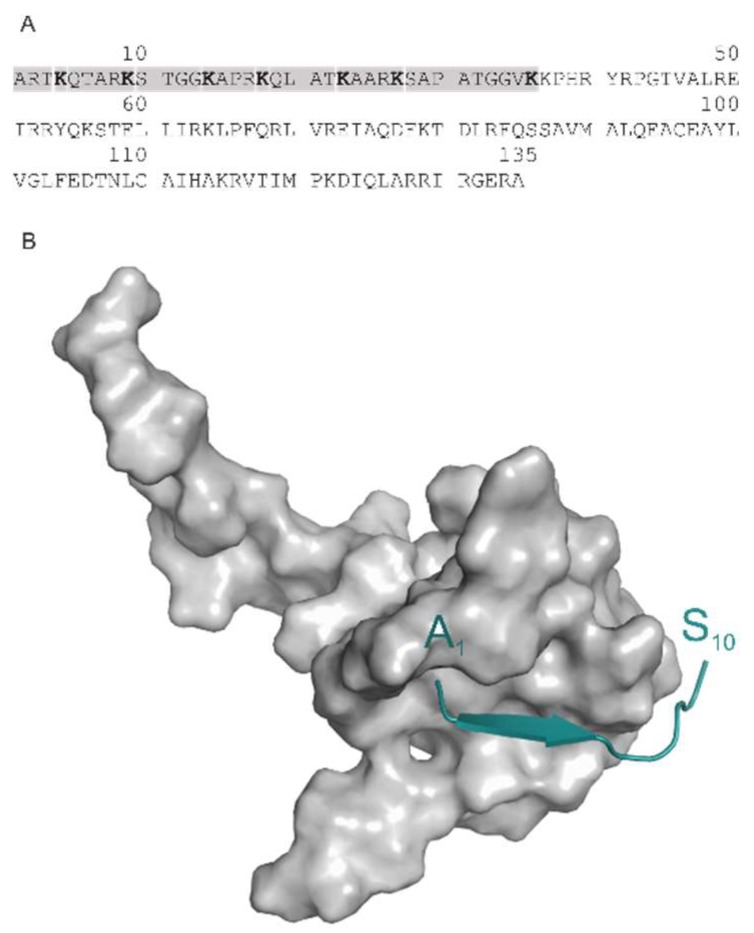
(**A**) The sequence of H3 with the 36-amino-acid long N-terminal peptide marked in grey. (**B**) The complex of the H3 decapeptide tail (green) bound to AIRE-PHD (System 2ke1) test case.

**Figure 2 ijms-20-00422-f002:**
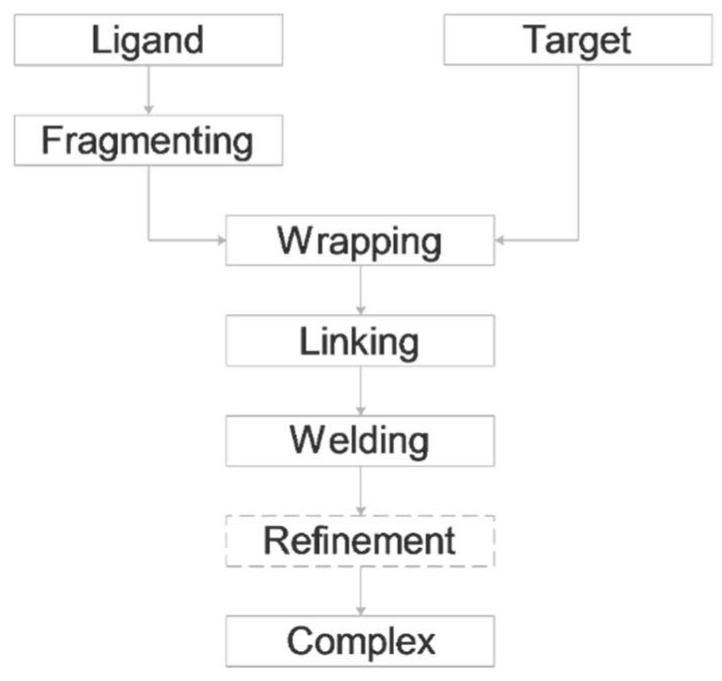
Fragment blind docking (FBD).

**Figure 3 ijms-20-00422-f003:**
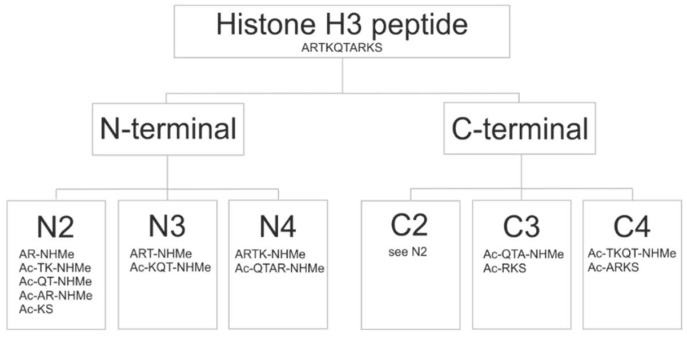
The fragmenting scheme.

**Figure 4 ijms-20-00422-f004:**
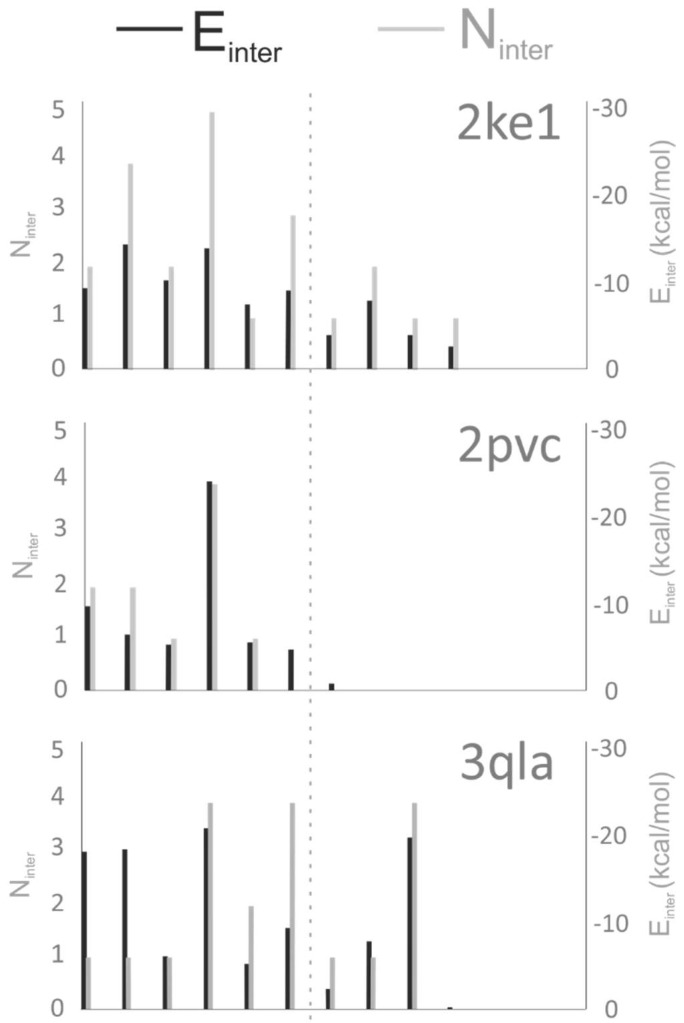
Per-residue values of target–ligand intermolecular interaction energy (E_inter_) and number of intermolecular interactions (N_inter_) calculated for the first 13 amino acids of the histone H3 ligand. The dotted line represents an approximate border of the N-terminal anchoring region where most of the target–ligand interactions act.

**Figure 5 ijms-20-00422-f005:**
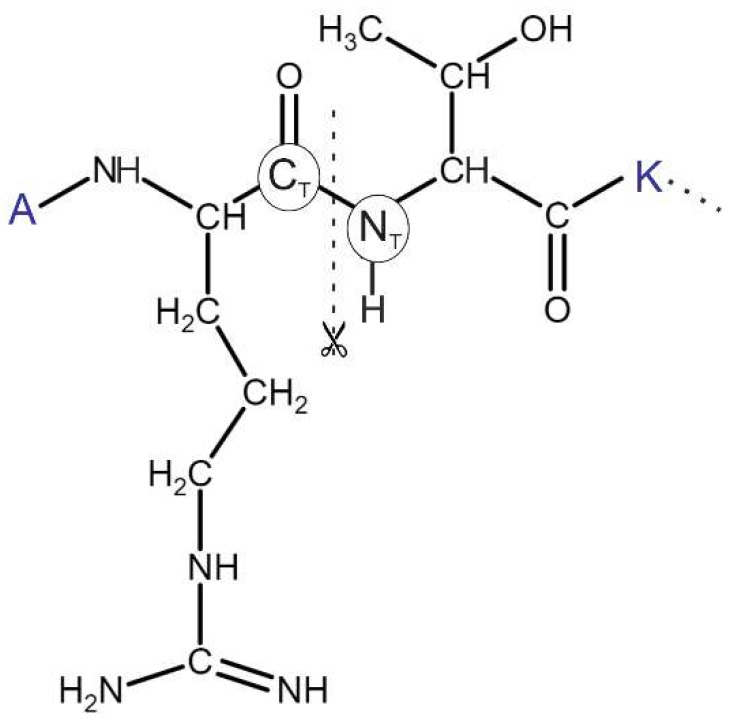
Chemical formula of the N-terminus (ARTK) of histone H3 with a detailed Lewis structure of R_2_ and T_3_. The peptide was cut into di-peptide fragments between the C_T_ and N_T_ atoms of the amide bond.

**Figure 6 ijms-20-00422-f006:**
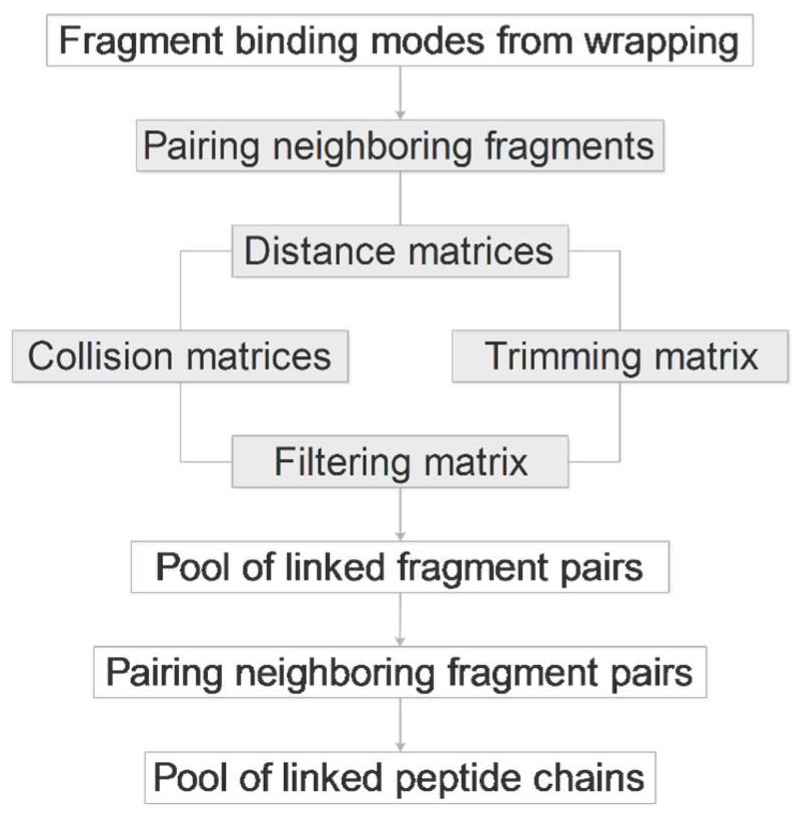
The linking algorithm (grey boxes refer to repeated tasks for all fragments).

**Figure 7 ijms-20-00422-f007:**
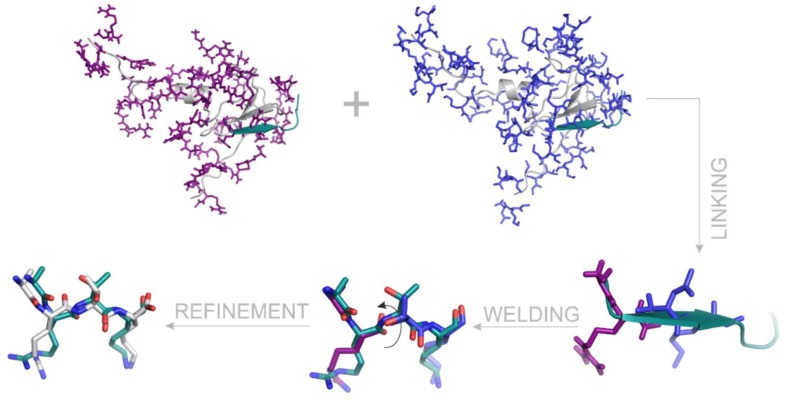
The process of linking, welding and refinement represented on the example of a fragment pair AR-NHMe and Ac-TK-NHMe derived from the histone H3 peptide ligand of System 2ke1. For comparison, the crystallographic ligand conformation is represented in teal cartoon on the first tree images, and with teal sticks on the last two images. The AIRE PHD target is in grey cartoon. Calculated binding modes of AR-NHMe and Ac-TK-NHMe after wrapping of the target are represented with purple and blue sticks, respectively. After linking, binding modes matching with the crystallographic H3 conformation are shown. During welding, the capping groups are removed and the distance between the terminal (C_T_ and N_T_) atoms is minimized via intra-molecular rotation (arrow). Refinement with molecular mechanics minimization helps formation of the proper bound structure of the ARTK fragment (grey sticks).

**Table 1 ijms-20-00422-t001:** Test systems.

PDB ID	Target	Ligand (Histone H3 Peptide) *
2ke1	autoimmune regulator protein plant homeodomain (AIRE-PHD)	ARTKQTARKS
2pvc	DNA (cytosine-5)-methyltransferase 3-like (DNMT3L)	ARTKQTA
3qla	transcriptional regulator ATRX-ADD domain (ATRX-DNMT3-DNMT3L)	ARTKQTARK(Me_3_)S
4lk9	histone acetyltransferase KAT6A	ARTKQTARKSTGG
5tdw	Set domain containing protein 3	ARTK(Me_3_)QTARKST

* One letter amino acid codes are used with PTMs marked in brackets after the modified amino acids.

**Table 2 ijms-20-00422-t002:** Docking results *.

System	Fragment Type	Fragment Sequence	#Cycle	#Rank	RMSD (Å)
2ke1	N2 = C2	AR-NHMe	1	4	1.9
	N2 = C2	Ac-TK-NHMe	1	1	2.8
	N3	Ac-KQT-NHMe	1	1	2.4
2pvc	N2	AR-NHMe	2	5	3.2
	N2	Ac-TK-NHMe	1	3	2.8
	N3	Ac-KQTA	1	1	3.0
	C2	Ac-KQ-NHMe	1	2	1.7
3qla	N2 = C2	Ac-QT-NHMe	1	12	2.0
	N2 = C2	Ac-K(Me_3_)S	1	2	3.7
	N4	ARTK-NHMe	1	1	3.3
4lk9	N2	AR-NHMe	1	1	2.9
	N2	Ac-TK-NHMe	1	4	4.0
5tdw	N2	AR-NHMe	1	1	1.7

* Structures are shown in Appendix A. ^#^ Serial number.

**Table 3 ijms-20-00422-t003:** Matrices used during the linking process.

Symbol	Description
d_all_	Smallest distance calculated between all heavy atoms
c_all_	Collision matrix between any heavy atoms
d_CN_	Distance calculated between C_T_ and N_T_ atoms
c_CN_	Collision matrix from d_CN_, c_CN_ = 1 if there is no collision, otherwise 0
t_CN_	Trimming matrix from d_CN_, t_CN_ = 0 if trimmed, otherwise 1
f_CN_	Filtering matrix from the collision and trimming matrices, f_CN_ = 1 if c_CN_ = t_CN_ = 1, otherwise 0

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
