# Peer review of "Towards Unraveling the Histone Code by Fragment Blind Docking"

_ijms, 2019, doi:10.3390/ijms20020422_

Round 1

Reviewer 1 Report

I find the manuscript interesting and do not have any specific concerns. I have a suggestion. The predicted models for all the test cases can be uploaded to the online digital repository (e.g. figshare) and share their research outputs to the readers.

Author Response

I find the manuscript interesting and do not have any specific concerns. I have a suggestion. The predicted models for all the test cases can be uploaded to the online digital repository (e.g. figshare) and share their research outputs to the readers.

Response. We acknowledge Reviewer 1 for careful evaluation of our manuscript and also for his/her suggestion of sharing the predicted models in figshare at the online digital repository. Accordingly, a set of figures on the predicted models were prepared as listed in Table 2.

Reviewer 2 Report

This manuscript falls within the scope of  the International Journal of Molecular Sciences. The authors attempted to combine fragment docking and blind docking approaches and apply to determine the structures of histone-reader complexes, which is a challenging problem from X-ray or NMR due to the large size of PTM variations.

However, the test systems used in the manuscript are limited to highly similarly ligands. Major revision is required before considering for publication.

1. As a new methodology, what's the major advantage and limitation of FBD comparing to other approaches?

2. Considering the most common PTMs, such as methylation, acetylation, phosphorylation. The authors only looked at the tri-methylated PTMs. The number of test systems are too few to make a convincing conclusion. The method may be biased to specific sequence patterns or charge pattern.

3. What's the rank of peptide fragments by just using the native ligand binding conformations? If the rank is low, what's the possible way to improve the scoring and generate more accurate docking poses?

Author Response

This manuscript falls within the scope of  the International Journal of Molecular Sciences. The authors attempted to combine fragment docking and blind docking approaches and apply to determine the structures of histone-reader complexes, which is a challenging problem from X-ray or NMR due to the large size of PTM variations.

However, the test systems used in the manuscript are limited to highly similarly ligands. Major revision is required before considering for publication.

Response. We acknowledge Reviewer 2 for careful evaluation of our manuscript. The manuscript is focused on the interactions of N-terminal histone peptides as ligands. As the sequence of histones is conserved within a subfamily (the H3.1 in our case), the sequences of peptide ligands derived are practically identical. The ligands are expected to have differences only in their lengths or in their PTMs. However, the peptide ligands of the present study are involved in various interactions with different target proteins. At the same time, from a method development viewpoint it was beneficial to investigate the docking of sets of similar peptides as systematic observations could be collected on per-residue interactions (e.g. in Fig. 4) which largely helped drawing relevant conclusions.

1. As a new methodology, what's the major advantage and limitation of FBD comparing to other approaches?

Response to point 1. Heuristic search engines of present fast docking methods cannot handle peptide ligands with numerous internal rotations. To handle this situation, fragmenting has been used in previous fast docking studies focusing on a known binding pocket. The present study provided the first application of fragmenting in a blind docking context with no restriction of the search space. Thus, even the approximate knowledge of location of binding pocket was not necessary in our successful examples. Section Conclusions is extended with the above text on comparisons (marked with yellow background).

2. Considering the most common PTMs, such as methylation, acetylation, phosphorylation. The authors only looked at the tri-methylated PTMs. The number of test systems are too few to make a convincing conclusion. The method may be biased to specific sequence patterns or charge pattern.

Response to point 2. The aim of the present study is the introduction of FBD for the difficult problem of histone-protein interactions. We checked the feasibility of the main ideas of the method. Extensive docking trials (>9000 runs per system) had to be performed for a systematic elaboration of the new methodology according to the fragmenting scheme as described in the first paragraphs of Section Results and Discussion. While the number of test systems is not high, the conclusions of the manuscript are solid, as they are based on the above-mentioned numerous trials (computational experiments). In accordance with the above remark of Reviewer 2, the last statement of Section Conclusions was modified to help the reader in appropriate evaluation of our new findings (marked with yellow background).

3. What's the rank of peptide fragments by just using the native ligand binding conformations? If the rank is low, what's the possible way to improve the scoring and generate more accurate docking poses?

Response to point 3. The rank of peptide fragments with the best structural fit to the native ligand binding conformations are shown in Column 5 of Table 2. In 12 of 13 cases, the docked fragments are placed in the top five ranks, half of which is Rank 1. Thus, the best structures have the best energy scores (top ranks) which is appropriate in such docking studies (Ref. 19).

Round 2

Reviewer 2 Report

All my questions are clearly answered. No further review is required.

The manuscript can be accepted and considered for publication.